# Geology Uprooted! Decolonising the Curriculum for Geologists.

Steven L. Rogers[1], Lisa Lau[1], Natasha Dowey [2], Hinna Sheikh[3], Rebecca Williams[4]

[1]The School of Geography, Geology and the Environment. Keele University, UK.
[2]Department of the Natural and Built Environment, Sheffield Hallam University, UK
[3]Race Equality Officer. Keele University, UK.
[4]Department of Geography, Geology & Environment, University of Hull, UK.

*Correspondence to*: Steven L. Rogers (s.l.rogers@keele.ac.uk)

**Abstract.** Geology is colonial. It has a colonial past, and a colonial present. Most of the knowledge we accept as the modern
discipline of geology was founded during the height of the post-1700 European Empire's colonial expansion. Knowledge is not neutral, and its creation and use can be damaging to individuals and peoples. The concept of Decolonising the Curriculum has gathered attention recently, but this concept can be misunderstood or difficult to engage with by individuals who are not familiar, or trained to work, with the literature on the issue. This paper aims to demystify Decolonising the Curriculum, particularly with respect to geology. We explain what Decolonising the Curriculum is, and outline frameworks
and terminology often found in decolonising literature. We discuss how geology is based on colonised knowledge and what effects this may have. We explore how we might decolonise the subject and most importantly, why it matters. Together, through collaborative networks, we need to decolonise geology to ensure our discipline is inclusive, accessible to all and relevant to the grand challenges facing diverse world societies.

## 1 Introduction

Decolonising the Curriculum is an initiative that has gained momentum around the globe in recent years (Charles, 2019). Its origins are in Humanities and Social Sciences and therefore some of the language and rhetoric used, issues raised, and supporting texts, experiences, theories and ideas may be impenetrable or unfamiliar to those from STEM backgrounds. As academics there is good evidence that we most comfortably operate in 'discipline silos' of individuals who we feel share common interests, values and skills (Becher and Trowler, 2001; Amaral, 2008; Kreber, 2008 and Rogers and Cage, 2017). It
is understandable, therefore, that STEM groups may lack the expertise to unravel some of the scholarly work around Decolonising the Curriculum. It is also true that many geoscience departments lack pedagogic experts (this is particularly true in the UK). This piece aims to break down some of the barriers to accessing and understanding Decolonising the Curriculum, and in this case is framed around the discipline of geology (however it should be of use to academics across STEM disciplines). We have tried to avoid language that might be unfamiliar to geologists and have provided a glossary of
words and phrases that commonly appear in scholarly work on Decolonising the Curriculum and pedagogy. Examples of

colonial geological legacy are given, and we explore how and why this legacy may be problematic. We also suggest ways in which Decolonising the Curriculum can make our discipline more open, accessible, modern and inclusive.

We openly acknowledge that this piece does not fully delve into every specific of the geology curricula or provide explicit 'fixes' – this is very much designed to explain what Decolonising the Curriculum is, particularly where geology is concerned, and some ideas of how to approach Decolonising the Curriculum are provided. This is an introduction to be built on. It is intended to demystify Decolonising the Curriculum and show its applicability to geology and geologists. Future work by the authors and collaborators will involve exploring local and Indigenous people's geological knowledge and their role in colonial geological surveys in the late 19[th] and early 20[th] centuries, investigating the colonial present of geology, and

developing open-access resources for a decolonised geology curriculum.

The authors of this paper all work at universities in the UK and are actively involved in exploring, leading, and/or promoting/explaining the Decolonising the Curriculum initiative at their institutes. Levels of engagement and experience with the Decolonising initiative varies amongst the authors; from programme level involvement to institute wide

responsibility and growing national recognition. Of the authors, three are geologists/geoscientists, one is a postcolonialist from Malaysia and one is a Race Equality Officer. The authors are UK-based, and we recognise the privilege we have as scholars of the Global North, but we equally uphold that it is imperative that decolonisation efforts must happen from colonising countries.

This piece focuses on Decolonising the Curriculum, but it is important to emphasise that decolonialism is a much wider issue than the curriculum. It is and should be uncomfortable. This piece is very much for the academics in the Global North whose curricula had in the past been a relatively colonised curricula (as per tradition) but who are trying to find out how they can decolonise their teaching and learning in order to provide a more inclusive, representative curricula. Tuck and Yang (2012) emphasise that calling for decolonisation (of schools, for example) turns it into a metaphor where it is used as a vehicle for

social justice and other methodologies – but not decolonisation. Our paper hopes to encourage decolonisation as more than a metaphor, but our focus is on knowledge and learning, not on the entire broad swath that decolonisation can encompass. Our focus is on how the logic of coloniality (Mignolo 2007) has pervaded in Western Modern Science and in the knowledge construction within Western (as well as non-Western) universities, and how we can take steps to make a paradigm shift, and break the shackles of colonised minds, curricula, and knowledge sets. We start out with baby steps, first to raise awareness of

the problem, and in future publications we hope to be able to identify strategies that would be of relevance to geologists.

## 2 The foundations and dominance of colonial geological knowledge

As an academic discipline or branch of knowledge, geology is relatively young (the academic discipline of geology arose in Europe - and to an extent the United States of America - in the late 18th/early 19th century). There are references to geological knowledge in several ancient texts (including the creation/formation of certain rock formations and the links to ancient environments, ideas on plate tectonics, etc.), mostly attributed to polymaths from around the globe or by scholars of 'other' subjects and theologians (e.g., Theophrastus, BC 371-287, an Ancient Greek philosopher (Cuvier, 1830); Pliny the Elder, AD 23/4-79, an Ancient Roman natural philosopher (Pliny the Elder, 1855); Abu al-Rayla al Birun, AD 973-1048, an Iranian scholar (Asimov and Bedworth, 1998); and Shen Kuo or Shen Gua, courtesy name Cunzhong and pseudonym Mengqi/Mengxi Weng, AD 1031-1095, a Chinese polymath (Yao, 2003)). However, the study of the Earth and its changes through time has only really developed as a distinct academic pursuit since the late 18th century, arguably driven by a mixture of advanced mobility (the ability for individuals to cross vast distances recording rocks and relating them to one another), resource exploitation, and an increased interest in understanding 'what' Earth and its constituent systems are. The first two of these motivating factors have strong colonial roots; it was at the height of colonial Europe that many of the principles, theories, laws and practices that shape the discipline of geology were established. Prior to the late 18th century, the economically/resource-driven activities, the colonial Spanish and Portuguese state-led mineral exploitation of the early 16th century (Studnicki-Gizbert and Schecter, 2010), for example, that we might include under the broad umbrella of geology today (i.e. surveying, quarrying and mining) cannot be considered academic in nature (Sangwan, 1993). Although these early accounts of mineral exploitation are not considered academic, these activities did pave the way for further expansion and colonisation, and ultimately contributed to the mindset that Empires had the right to survey for, extract and trade mineral wealth – this process laying the foundations of the modern discipline.

The principles and practices established in the early formation of the discipline were made (and/or sponsored) by men who were privileged (mostly wealthy) enough to pursue academic interests, both in their native countries (nearly exclusively European) and increasingly across borders. Ultimately, the discipline of geology as we know it was born at the height of Northern European Empires (with the foundations very much provided by earlier periods of Southern European Empire expansion and exploitation). But what does that mean for the subject - what difference does it make?

The Global North's (Eurocentric) dominance of knowledge production (i.e. epistemology – see glossary) has, as Winkler (2018) observed, led academic disciplines born of colonialism to "the tendency to systematically classify philosophical concepts for the purpose of organising knowledge into distinct properties [which] has become a hallmark of Western scientific reason." (p592). The foundations of geology have been built on these pedagogical limitations. A dominant epistemology informs what knowledge a discipline is built from, and how it is taught (i.e. pedagogy - see glossary). Decolonising the Curriculum seeks to explore and question the epistemology of a discipline and looks to reform it (what is important knowledge?) and in doing so influences and alters a discipline's pedagogy.

Carey et al. (2016) have pointed out that the modern (see 'Baconian' in the Glossary) view of knowledge creation (modern scientific method) "engendered a strong tendency in the environmental sciences to classify, measure, map, and, ideally, dominate and control nonhuman nature as if it were a knowable and predictable machine, rather than dynamic, chaotic, unpredictable, and coupled natural-human systems" (p777). Rudolph et al. (2018) explain western universities' dominance in knowledge construction, production and legitimisation. They explain how 'powerful knowledge' is produced and refined in specialisations, predominantly in resource-rich universities, predominantly in the Global North. These institutions play by a set of internalised structures and hierarchies and acknowledge internal rules, which go towards reinforcing colonial and racist power relations. Such 'powerful knowledge' continues to ignore, belittle, and erase other systems of knowledge. These structures also make institutes and universities potentially hostile environments for Indigenous scholars (staff and students) who must conform to dominant systems and suppress their Indigenous knowledge and identity (Dzombak, 2020). What Peake and Kobayashi (2002) said of the discipline of Geography – that "Without an explicit effort being made to address and correct the consequences of the various (and often hidden) racist practices and discourses that permeate the epistemological foundations of geography and the institutional structures and practices that shape our work environment, geography will continue to embrace the colonialist heritage bequeathed upon it" (p50). – is also true of geology and other STEM disciplines.

Colonisation of knowledge still takes place today and whilst it is the legacy of historical colonial powers, it is not limited to their activities; it happens through imbalanced power relationships, internally within societies as well as externally (Popperl, 2018; Turner, 2018 and Calvert, 2001). The colonisation of knowledge is engineered by proclaimed experts, whose power typically originates from an elite (generally wealthy) societal group or standing. We recognise that calling for a decolonised curriculum is not just permitting inclusion of more viewpoints or amplifying the voices of a wider range of experts; the power imbalances are systemic, structural, and at all levels, particularly institutional, Trisos et al. (2021) highlight how difficult reconstructing power hierarchies will be: "Particularly for white-bodied researchers at well-funded universities and other organizations funded by corporate wealth from resource extraction, giving up a power and voice that has been explicitly and implicitly reinforced for at least 500 years will not be easy." p1209).

Colonisation of knowledge can happen where governments or corporations are involved, where organisations or institutions set the norms, but can also happen on a personal level, for example in the power balance between students and their tutors/supervisors. In geology (and many other disciplines) some examples of modern colonialism of knowledge include the influence of world powers, scientific associations and societies and publishing houses. The influence of these groups differs, but ultimately, they have control over processes that result in the promotion (plus the extraction and funding) of knowledge seen as valuable to them. "English has been the dominant form of knowledge communication in science, which can lead to publication bias against non-native English-speaking scientists. When one reads, writes and thinks in English, it is easy to forget that for the majority of people […] knowledge is produced and tested in other tongues" (Trisos et al., 2021, p1206)

Many readers will have experienced some level of this; peer review of papers and grant applications, and the biases involved, is a ubiquitous power structure that can determine which knowledge might be seen/perceived as valuable. Regulatory and/or accrediting bodies for education are also modern examples where a small group of individuals have the power to dictate knowledge that is valuable to a discipline. Indigenous scholars in geology/geoscience are (like all historically excluded groups) poorly represented within the discipline, and struggle to be accepted.

## 3 What is Decolonising the Curriculum?

### 3.1 Origins and overview

Decolonising the Curriculum is an initiative focussing on the action of decolonising ourselves (students and staff) and the teaching environments in which we operate. The founding frameworks of this movement are generally agreed to be the areas of decolonial and postcolonial studies. Postcolonial studies tend to focus on the social, economic, political, and cultural impact of colonial powers on past colonies and their peoples. Decolonial scholars arise from a variety of disciplines and are generally concerned with epistemology – questioning the dominance of Eurocentric knowledge systems. Both decolonial and postcolonial studies have been around for several decades and are generally studied by Indigenous scholars. Decolonising the Curriculum is at different stages around the globe. Postcolonial nations (nations that were once colonies) are leading the way with several examples of Indigenous knowledge systems or methodologies being embedded within curricula; Manathunga (2020), for example, provides several vivid examples from New Zealand and Australia.

Although there is no single definition or understanding of what a curriculum is (Egan, 1978; Young, 2014), a curriculum can generally be described as the total sum of the knowledge, skills, social norms and experiences that a student learns, or are exposed to, within a designed educational process. A curriculum is formed of/based on the knowledge, skills and experiences considered to be valuable for a discipline, and certainly for geology, any industry that has influenced it.

Decolonising the Curriculum is not an initiative looking to shame individuals for the content they teach, or for the work they make use of. It is not about cherry picking diverse content for the sake of diversity or deleting certain works, and is not an outright ban on teaching the work of old, dead, white men (Pett, 2015). It is not about change for the sake of change, and it is not about the formerly colonised and the colonised switching places either. Decolonising the Curriculum recognises that a total or outright dismantling or destruction of all imperially created structures and processes is not likely to happen overnight, nor without resistance.

However, a restructuring would be helpful; a rebalancing of power to decrease the marginalisation and othering (see Glossary) of groups and knowledges could result in better pedagogy and greater understanding. It would also aid inclusivity through improved representation and diversity. Decolonising the Curriculum has also been highlighted as a method towards closing the degree awarding gap between white students and those from a Black, Asian or Minority Ethnic background (UUK and NUS, 2019). So, while Decolonising the Curriculum does not call for the abandonment of all Western theory, it does flag up that Western theory "does not in fact describe or map the entire planet, and that despite pretensions to

universalism it suffers from gaps and lacunae, and for this reason needs to be revised in the light of local empirical conditions" (Jackson, 2003, p73; in Hönke and Müller, 2012, p390).


### 3.2 Envisioning all cultures and knowledges

There are several definitions of decolonisation and Decolonising the Curriculum. Mbembe (2016) highlights that there is little agreement on what decolonisation is and that it is an everchanging and evolving "Beast". Here, as did Charles (2019), we support this definition taken from part of Keele University's Decolonising the Curriculum Manifesto (Keele University,

165    2018):

"Decolonising the curriculum means creating spaces and resources for a dialogue among all members of the university on how to imagine and envision all cultures and knowledge systems in the curriculum, and with respect to what is being taught and how it frames the world."

Decolonising the Curriculum is a philosophical and pedagogical initiative exploring the origin, development and use of

knowledge, looking not only at repositioning theory, but also the content of a curriculum, and how that content is taught. It is a curriculum design process looking to recognise knowledge as power, as well as recognising the power that enabled knowledge to be legitimised as such. It encourages us to question who created certain knowledges, why we use that particular knowledge, and who has access to it and why. A decolonised curriculum explores and acknowledges colonial legacy in knowledge creation, giving credit to those hidden and minoritised individuals who deserve it. Decolonising the

Curriculum is about exploring, examining, interrogating, and teaching the history of a discipline's knowledgebase. It involves inquiring about the approach, method, framing, thought paradigms, theories, structures and concepts that underpin and form all content within the discipline. The initiative is not solely concerned with knowledge, however. It is also vitally about place, power, and identity. Many Indigenous scholars, including geoscientists, have commented on how academics and students often have to assimilate into academia, following the norms, structures, frameworks, behaviours and knowledge

systems imposed on them. Dzombak (2020) provides a short blog highlighting the experiences of a few Indigenous geoscientists.

### 3.3 A reflective, uncomfortable process

Decolonising the Curriculum is sure to mean different things to different people and will involve different actions for

different disciplines. This seems particularly true if we compare subjects from STEM, the Social Sciences and Humanities. The process requires us to reflect on our backgrounds, experiences, ideologies and discipline-specific narratives. Drawing on Tuck and Yang's (2012) work calling for the non-domestication of decolonisation, Esson et al. (2017) eloquently argue that "Decolonisation is a radical challenge to 'unsettle' the architecture of privilege" (p387). As academics who occupy positions of privilege and are sometimes said to dwell in ivory towers, decolonising our curriculums has to be a deeply self-reflexive

process involving capturing the experiences of historically marginalized groups (decolonising aims to address and rebalance

injustices for all marginalised groups – it is not to be mistaken solely as a race/ethnicity issue). We need to acknowledge the biases in our world views (including social and political) (Holmes, 2020), be aware of our relationship to curricula/research and fully understand ourselves as educators and researchers, to address the context in which curricula design (both in terms of content but also in terms of practice) (Rose, 1997) is taking place. To some extent, this is a complex manoeuvre where we

sometimes have to pull the rug out from under our own feet. And while decolonisation of anything, let alone curricula, is clearly many pronged, multi-levelled, and complex, one thing it definitely will also be is discomfiting; those who undertake to decolonise must be prepared to step outside of comfort zones and interrogate assumptions and privileges, and perhaps even unlearn some of the latter (as Spivak advocates; Spivak, 1990 and Andreotti, 2007).

Decolonising the Curriculum should ideally be a reflective and honest process where we recognise the emergence and use of

the knowledge, or set(s) of knowledge, we choose to apply in any given circumstance. Under what circumstances was the knowledge we use made, and why do we use this set of knowledge in particular? Several authors outline what decolonisation might entail, and include themes such as recovering knowledge, reflecting on the exclusion of other knowledge, ethics, the use of language and the internationalisation of Indigenous experience (e.g. Smith, 1999; Chilisa, 2020; Le Grange, 2020).

Whilst Decolonising the Curriculum is not a call for the vilification of past individuals this does not mean we cannot judge

and be disappointed, embarrassed or angry at some of the unjust assumptions, beliefs and actions of figures in the past. It is a call to understand, reflect and call out the norms and actions of those who provided important advances to our knowledge. How might those behaviours have arisen, and how did they impact the formation of the knowledge we use? Were there others involved in the formation of that knowledge, who were forgotten or marginalised collaborators? How might that have led to the continued exclusion of some groups in the present? The process acknowledges how certain knowledge was

created, for example, by explaining where authors held views which are found repugnant today, or where data was gained at the expense of others. For example, Lam (2021) provides an informative piece outlining (amongst other colonial links to geosciences) the history of Henry De La Beche, a slave owner who advocated for slavery reforms rather than abolition (De La Beche, 1825) and who is credited for creating the first geological map of Jamaica whilst visiting his slave plantation.

Decolonisation calls for us to consider the broader pool of knowledge available outside of our Eurocentric curricula (Hall

and Tandon, 2017). Teaching geology from a single perspective (often framed by the works of dead white men) leads to uneven power relations particularly in relation to race, class and gender (Begum and Saini, 2019). Where knowledge is ignored, or ownership of knowledge is denied as to originating or belonging to Indigenous peoples, damage and hurt can be caused (Whitt, 2009). What we say and do matters.

**3.4 Bridging society and science**

Importantly, Decolonising the Curriculum seeks to highlight how injustice in the past has led to some of the most fundamental aspects of modern thinking, discipline identities and continuing inequities (Harding, 2006) – and how by acknowledging and understanding this, we can be better and strive to make a just, fair, equitable and accessible modern

system that provides a curriculum relevant to modern challenges. An issue not just for geology but all STEM subjects is that science is often presented as being apolitical or neutral, where social relations have little bearing or influence. Gracio (2014) highlights this issue - "Science is made by people with interests, intentions and ambitions; and it's funded by governments and companies with agendas." - and calls out the absence of ethics and politics teaching in science curricula. The idea of "political geology", an inter/multi/trans-disciplinary area relating to geopolitics, the Anthropocene, technology, cartography, history and other themes has been identified as an emerging area of work (Bobbette and Donovan, 2019).

## 3.5 A holistic process- beyond diversification

Some of the conversations around Decolonising the Curriculum focus (in an unnecessarily constrained and limited way) solely on diversifying reading lists and case studies used across educational units. Diversity of representation in reading lists is important; ensuring reading lists are not just from Western male perspectives can enrich content and open the door to different frames of knowledge, experiences, and points of view. However, piecemeal developments like diversifying reading lists, whilst useful, do not fulfil the scope of Decolonising the Curriculum. Decolonising the Curriculum is a holistic process. It needs time, thought, collaboration and willingness, to not only take fragmentary steps but for a major overhaul. A common criticism of Decolonising the Curriculum is that it "removes" or conveniently effaces historical knowledge (by removing certain case studies, authors or contexts, for example). Done properly, it should in fact broaden frames of reference, recognising other knowledge systems and ways of thinking, and opening global dialogue. Vandeyar (2020, p5) provides a useful quote emphasising how we must go beyond the diversification of materials and ensure we challenge and interrogate the knowledge we use: "Decolonisation of the curriculum requires much more than just changing the curriculum. How things are taught and academics' attitudes, perceptions and beliefs in this process are pivotal to the decolonisation project. Decolonisation is more than just a 'choice of materials' (Wa Thiong'o, 1992). The attitude and disposition to materials used in the curriculum is critical."

## 4 Colonised Geology

### 4.1 Colonised forms of geological knowledge

Geology is a discipline created by colonial forces/parties at a time of active (explicit) colonial expansion (Yusoff, 2018; Figueiredo, 2020; Zeller, 2000). Because of its global relevance and common use of international case studies, it might be felt by some that geology is no longer colonial, or that the colonial roots of geology no longer influence the subject's arena. However, the discipline born during Imperial expansion is still very much the discipline taught across Western institutions today (albeit with adaptations as technologies/methods/nomenclatures/schemas have developed). It is important to recognise that this colonial version of geology, known to most geologists as the accepted global norm or canon (and adopted by many non-western countries likely as a result of colonial legacy), is not the only form of geological knowledge practiced today or

in the past. What makes the geological knowledge of a range of groups incompatible with the accepted geological canon? Many of the core aspects of geological teaching and learning focus on the identification, classification and physical/mechanical characteristics of Earth materials, echoing the geological activities promoted during the rise of geology as a military science and latterly an academic pursuit (see 4.2 Origins and 'firsts', below).

Many Indigenous peoples have described and used their local geology for thousands of years (Nyblade and McDonald, 2021 and references therein). Reano and Ridgway (2015) highlight some of the geological workings of the Acoma People (west-central New Mexico), who, rather than use the stratigraphic framework and classifications familiar to institutional geology (education, academia, industry) use an interpretive framework passed down generation to generation (called a "cultural framework" in their paper). This framework groups lithologies by their cultural or resource significance (e.g. farmland, building materials, pottery materials, water resources). These alternative frameworks can be linked and compared to "standard" frameworks to better welcome minority groups into geology (Reano and Ridgway, 2015). For wider cohorts, cultural frameworks also encourage better understanding of world views and the relationship between Indigenous populations and their lands, and highlight how cultural tensions can arise from modern colonisation (resource exploitation on Indigenous lands, for example).

There are many more examples of traditional and local knowledge, oral histories and mythology that are dismissed or ignored (and/or belittled) by the Western knowledge system but are grounded in the truth of observation. Oral histories include details of past environmental and climate change, of cataclysms and the resulting environmental response. Oral histories are cumulative through generations and can often cover large parts of human (and non-human – animals, plants, rivers etc.) history. Whilst Western scientific method can (unquestionably) answer many questions, decolonisation of what counts as knowledge is needed to integrate Indigenous knowledge systems. Nunn (2018) demonstrates this excellently by bringing together the knowledge systems of Western climate science canon and the oral histories of the Aboriginal First Nations peoples. For future climate action this sort of integrated dialogue will be invaluable – for example, we could potentially predict the effects of sea-level rise to a coastline whilst integrating knowledge of the environmental impact (including how animals and plants react to such events) as evidenced through historical knowledge.

Most UK/Eurocentric/Western geologists probably do not think that cultural/Indigenous/alternative frameworks of knowledge even exist locally – often regarding the local knowledge as 'less developed' (a colonial attitude!) - but that is far from the truth. Within the Global North, some parts which were themselves formerly colonial powers as well as internally colonised, have (or had) a wealth of unacknowledged local geological knowledge, some of which persists. For example, in South Shropshire (England) many locals refer to "dhustone" for a hard, black igneous rock quarried from a place called Clee Hill. If asked about dhustone, many locals would likely be able to tell you where this rock can be found, and why it is quarried. If, however, you were to ask locals if they knew where you could find micro-gabbro in the area, they would likely not know. This sort of geological knowledge, which exists across the globe, is often downplayed, or explained as 'not proper' geology – why? The knowledge serves a purpose and is successfully disseminated. Many of the terms used locally

for rock types with decorative aspects (such as Cotham marble, Purbeck marble, Sussex marble, and Puddingstone in the UK) are often dismissed as 'incorrect' by geologists. This narrow acceptance of what is 'correct' geological knowledge potentially damages the image of the geological discipline, with individuals being made to feel inferior and therefore unwilling to engage further. Learning about and working with local knowledge is not an onerous task and could lead to a more engaged and responsive reaction to geological activities (e.g. Palmer et al., 2009).

The connection between the geoscience industry and active harm to sites of cultural significance is a tangible result of the erasure and belittlement (or wilful misunderstanding and ignoring) of local and Indigenous knowledge. Geologists working for the extractive industry can hold power over the future of landscapes and peoples that have coexisted for thousands of years. Colonial legacy of land ownership and legalities over material extraction (supported by powerful and wealthy groups) is very much persistent today. The destruction of a 46,000-year-old First Nations heritage site of rock shelters in Western Australia to access higher volumes of high-grade ore is just one recent example (Wahlquist and Allam, 2020a). The responses to these inexcusable actions have been positive; all mining companies in Australia have been recommended to review all agreements with traditional landowners, and Rio Tinto have several recommendations imposed on them including remediation work, restitution packages and a commitment to halt actions on 1,700 First Nation heritage sites it has permission to destroy (Wahlquist and Allam, 2020b). Geoscience researchers have been responsible for similarly destructive activities, with several well documented cases of rock core/samples being taken from sites of cultural significance (e.g. Sahagún, 2021) and from areas of natural beauty (MacFadyen, 2010), in spite of codes of conduct existing to mitigate against this (e.g. Scottish National Heritage & the Geologists' Association, 2011).

Perhaps the most important aspect of acknowledging geology's colonial present, and its debts to marginalised peoples and damage to environments, is to ensure that present and future actions work towards a more collaborative discipline in which co-production of knowledge with all involved parties is normalised (Adame, 2021; Adams et al., 2014; Wilkinson et al., 2020; Sheffield et al., 2021). The geosciences are often overlooked (or misunderstood) in policy (Stow and Laming, 1991; Gill and Smith, 2021), process and considerations for sustainable development, and this is undoubtedly linked to past geological activities being associated with extractive and damaging processes.

## 4.2 Origins and 'firsts'

The history of the discipline of geology is rarely taught in detail on taught geology courses. Many of the "Fathers of Geology" and their achievements are included in teaching programmes, but little (or no) content centres around how the current discipline was formed. The colonial influence and exploitative actions underpinning the subject's foundations are not part of the discipline's canon (and neither is the discipline's colonial present). This section gives a very brief overview of the discipline's colonial origins. Some of this will be familiar to geologists, particularly the names of certain individuals and

their geological contribution. What may not be known by some geologists are the wider systems, actions, and processes that

these geological contributions were made under.

In the 17th century individuals such as Nicolas Steno began drawing up ideas about the deposition of sediments and the origin of fossils, questioning the accepted views of Earth science at the time (Adams, 1938; Gohau, 1990). The 18th century saw a realisation that minerals and ores (often inaccessible at the surface) could be found by studying certain natural phenomena. At this time two main schools of thought arose to explain the creation of Earth materials – Neptunism (also

called Diluvianism) and Plutonism. Neptunism argued that geological materials precipitated from water (much of this thinking was linked to Christian Bible teaching, particularly the great flood) (Gohau, 1990). Plutonists believed that volcanism was mainly responsible for rock formation – and alluded to the age of the Earth being very old and not understandable from the limited span of teachings from the Bible (Gohau, 1990). It was in the 19th century where the foundations of the discipline we know today (from Western education and industry) were founded. Uniformitarianism (and

the opposing catastrophism) was proposed by Charles Lyell in his 'Principles of Geology' (Lyell, 1830). Ideas around stratigraphic principles and relative dating began to be developed at this time.

In the 19th century geological investigation often included historical and ethnographic elements; geologists would investigate a wide variety of subjects including antiquities, ancestral and Indigenous myths and past civilization/human activity and used texts and oral history to investigate local geology (Chakrabarti, 2020). For example, Alexander von Humboldt, a German

polymath and geographer, integrated his experiences around the globe to try and explain/explore the distribution of a number of natural features (animals, plants, mountains, volcanoes etc.) through the measurement and recording of them on maps (Secord, 2018).

Geological expeditions/surveys (although not necessarily solely *geological* - many aspects of trade, botany and anthropology etc. were also included) were an instrumental tool of colonial expansion (e.g. Stafford, 1984 and 1988; Sangwan, 1993;

Yusoff, 2018; Figueiredo, 2020; Zeller, 2000). Expeditions and surveys played an important role in the economic, technological and cultural development of colonial powers (Britain and Spain in particular). Spanish engineers surveyed much of South America during 1750-1840s and British surveyors operated across the British Empire through the 1830s-70s (Teale, 1945; Chakrabarti, 2019; Stafford, 1984 and 1988; Miller, 2020). Many expeditions, surveys and 'missions' to countries and territories where colonies were later established included a geological element. Geological surveys were

undertaken and estimates of natural resources were made, with the colonial party often being guided by locals, these guides and the knowledge they shared was erased, 'rewritten' or taken without recognition, a system which can still be observed today (see 4.3 'Parachute' science below). Many cases of colonial expansion and occupation were based on the findings of these 'exploratory' parties, particularly where natural resources were involved (Stafford, 1988). Other reasons for colonial expansion included strategic military/trade locations (including the slave trade), areas for European settlers to live and the

desire to push colonial frontiers further into lands occupied by "savages" and "barbarians" (Webb, 2017). The importance of mineral wealth to the British Imperial effort was so commonly understood that military, naval and commercial (e.g. the East

India Company) officers were offered training to better equip them to make scientific observations and enquiry, with mineral wealth from the colonies permanently held on display in London (Stafford, 1984). Official (British) Geological Surveys (i.e organisations, rather than the action of surveying/exploring) were established in nearly all UK colonial territories from 1918 (Colonial Geological Surveys, 1944). At a similar time to European Colonial expansion a similar expansion of colonial settlers was occurring in the United States of America, where geology surveys evaluated economic value of land and drove expansion into resource rich areas (Nyblade and McDonald, 2021). As British colonialism and Empire was rising, in South America (Spanish America in particular) modern day nations were being established and territories being fought for. These nations had little or no access to Spanish mineral survey data, conducted in the territories by colonial expansionists (Miller, 2020). Miller (2020) argues that these fledgling nations (and ultimately all nations) are most closely defined by shared knowledge and knowledge systems – in the case of Spanish America knowledge exchange between Indigenous peoples and Spanish colonials had been ongoing for centuries.

Secord (2018) introduces the idea that during the time of colonial expansion (particularly North European Empires) – resource extraction was not the sole geological motivation. Geology, and the idea that the Earth held multiple 'lost worlds' and natural wonders became entangled with philosophy and literary works of the time, for example, Secord (2018) suggests that Conrad's *The Heart of Darkness* (1899) and Conan Doyle's *The Lost World* (1912) allowed readers to share and experience the thrill of exploration to wonderous lands with contemporary geologists.

Some of the leaders of exploratory parties are well known geologists and scientists today, with Humboldt, Darwin, De La Beche and Murchinson key players in the use of surveys for colonial expansion (Stafford, 1984; Secord, 2018). These surveys and the organisations responsible for them were funded by colonial and ruling powers, for example the Spanish Crown (e.g. funding expeditions to Peru, Chile, New Spain (Mexico) etc.)) and the British Crown and Government - often directly through military organisations (e.g. the Board of Ordinance) (Rose, 1996). Early geological activities of the British Empire had such strong military ties that for much of the 19th century, certainly in the UK, geology was perceived as a military science (Rose et al., 2019). Colonial geologists are responsible for the creation of most of the "first geological survey of ..[somewhere].." and are often associated with the first geological interpretations of the areas they surveyed. In some cases, they are attributed with the "first discovery" of mineral wealth, or of features they observed. This is, of course, absurd. Locals often provided valuable knowledge, guided and worked for these parties without formal credit or recognition, and were clearly aware of many geological features prior to their reported "discovery". For example, Frank Dixey, the first Director of the Directorate of Colonial Geological Surveys talks about "native information", carriers and escorts in his personal memoirs on surveying Sierra Leone (Dunham, 1983). This phenomenon is commonly known as 'firsting' (see Glossary). It was these types of activities that led to the establishment of the geological discipline we know today. In Spanish and Portuguese America, Indigenous populations were clearly successful mineral surveyors, as proven by the quantities of gold, silver and copper looted by the Spanish and Portuguese in the early 1500s. After this initial period of looting the colonial forces then surveyed and extracted vast amounts gold and silver (Bakewell, 1984). The colonial forces took with

them their survey data when withdrawing from colonies and the Indigenous peoples once again surveyed their lands for minerals (Miller, 2020).

Individuals such as De La Beche and Murchison were likely driven by the same excitement and inquisitiveness that many geologists share about how the world works. The "exciting" debates held by prominent geologists at the time concerning the establishment of geological periods was a factor in influencing MPs, noblemen, military officers and colonial administrators that geological knowledge and exploration could promote economic growth (Stafford, 1984). But many of these individuals were in the privileged position to pursue an academic lifestyle due to injustices towards others, both domestically and internationally (e.g. Hyde, 2020). The theme of lone geniuses making exciting discoveries, giving talks and moving specimens to research institutes/museums etc. persist today. Rarely (never?) is knowledge creation the achievement of the individual. The idea that knowledge is validated through certain associations or groups of individuals is also observable today – so whilst many of the individuals who were prominent in the establishment of geology as the discipline we know today are long gone, the systems which they formed and supported are still very much alive.

Of course, there are individuals in the history of geology who were advocates for justice; for example, William and Richard Phillips and William Allen, who were pivotal in the establishment of the Geological Society of London, were abolitionists (Lam, 2021). These individuals, however, were still part of a group that encouraged the removal of materials from colonial territories for "metropolitan analysis" (Stafford, 1984). Imperial resource extraction may seem like an action of the distant past; however, geology as an essential tool for colonial expansion was celebrated as recently as the 1940s and 50s (Teale, 1945), was a dominant economic process until relatively recently, and arguably still continues via modern corporations. Reports of mineral extraction from colonial territories and scientific work resulting from such activities were published in the "Bulletin of the Imperial Institute" and latterly in "the Quarterly Bulletin of the Colonial Geological Surveys" up to at least 1957 (Beard, 1950).

### 4.3 'Parachute' science

Parachute knowledge creation is a phenomenon not restricted to geology or STEM disciplines. It is the act of researchers (typically from the Global North) traveling to conduct fieldwork in a 'Majority World' region (typically the Global South) and either not collaborating with, or not recognising the participation of, local researchers, landowners, or guides (e.g. Greshko, 2020; North et al. 2020). Spivak argues that field data collection (which she refers to as "information retrieval") is another form of Imperialism, which centres the Western academy (Andreotti, 2007; Nordling, 2021).

Parachute knowledge creation may involve the removal of samples or specimens from countries to be held or exhibited elsewhere without full collaboration or agreement from the country/area/people of origin, often referred to as extractivism (see Glossary). Although long the norm and even common practice in academia, extractivism is inherently exploitative. It may lead to the creation of academic outputs (e.g. articles published in academic journals) where the authorship team is

exclusively from the Global North, and collaborators from the study area are not included or acknowledged. These extractive practices have been shown to lead to biases in data, and still occur today (Raja et al., 2022). This process leads to the perception of the need for external experts to local issues; it doesn't meet or help local research efforts and can even hinder these local efforts (Stefanoudis et al., 2021). The practice of hindering local efforts has been recently highlighted by local geologists working on Nyiragongo volcano, Democratic Republic of Congo (Nordling, 2021).

Parachute/colonial science often leads to the phenomenon of firsting (see Glossary). A recent example of this practice has been reasonably visible and involves a unique Brazilian fossil that ended up in a German museum and was subsequently published on by a group with no Brazillian collaborators (Vogel, 2020). The example of the Brazillian fossil also raised questions on the ethical (and legal) practices of obtaining materials; Brazilian law forbids the exportation of fossils, other than for loans (Vogel, 2020). It is important to recognise that these behaviours can cause hurt to those being othered, and result in the breakdown of engagement, trust and willingness to help from these parties. Cisneros et al. (2022) outline additional examples of parachute and extractive science from Mexico and Brazil, as well as outlining the impacts (and excuses) of such practice. They suggest that scientists should be required to provide documentation proving the ethical and legal position of sample collection/acquisition, and that journals should refuse to publish without these.

Yozwiak et al. (2016) highlight that international collaboration is fundamental to tackling major global health emergencies. This is also true for tackling geoscience-related challenges such as climate change, critical material extraction, disaster risk reduction, and water extraction. Equitable collaborations between global experts, including those with invaluable local knowledge, are essential to avoid the damage caused by colonial science. Building research collaborations with support, training and educational opportunities for local communities helps engage key stakeholders and creates more equitable partnerships (Whiteford and Vindrola-Padros, 2015). These collaborative actions may seem daunting to those without the experience, time, resources, or incentives to carry them out (Roldan-Hernandez et al., 2020), but they should be normalized and built into ethical planning and research grant submissions.

## 5 Towards a Decolonised Geology Curriculum.

In decolonising the geology curriculum, we need to acknowledge the colonial legacy of the knowledge we teach and understand that the knowledge sets we use are not superior, not truth-claims, not pluriversal or representative, and therefore can only be partial and non-exhaustive/comprehensive. We must recognise the damage and harm which that knowledge creation was, and is still, part of; that some knowledge has been suppressed, erased, and that some has been created unethically. Geology is not apolitical nor is it unconnected to the sustainable future of diverse societies. We need to understand that all knowledge used has power.

To create a discipline that is equitable, progressive, and compassionate, curriculum development teams need to start considering decolonisation of their curricula now. The process will take time, effort, and willingness. Sharing effective practice, collaboration, co-creation, and listening to individuals from colonised territories, or those whose knowledge has

been colonised, is vital. There will be a wide range of actions specific to different curricula, dependent on what, how and where it is taught - each journey will be unique. Here we outline some suggestions on how we can begin to decolonise the geology curriculum:

1. **Explain and explore what Decolonising the Curriculum is.** Invite students to participate. Create or share resources that help explain what decolonising means. Emphasise the focus on knowledge production and use, and of power in the process of knowledge generation and suppression. Outline that it is not good vs bad and not about removing bodies of work based on individual beliefs and behaviours, but about exploring how this has influenced both the knowledge itself, and how individuals were oppressed or disadvantaged during the knowledge creation process. Explore why we should learn from this history, rather than repeat it.

2. **Teach the history of geology.** No geology is neutral (Yusoff, 2018). Teaching this discipline needs to include pointing to its framing. Exploring the origins of the knowledge we use and acknowledging that peoples and lands were damaged in the creation of that knowledge, allows us to understand why some groups might feel they do not belong in geology and how some groups have been excluded. It may help explain why diversity in geology cohorts is worryingly low (Dowey et al., 2021). It allows us and our students to understand the consequences of past actions and hopefully reduce/remove these actions in the present and future.

3. **Set the Context of the Discipline of Geology.** Instead of presenting the syllabi or curriculum as the definitive, universal version of 'Geology', contextualise to make clear that the geology taught in our degree programs is one version of many possible knowledges, from particular perspectives, and that it is selective and exclusive in various ways, as all curricula must be ('GeoContext', Pico et al., 2021). Make clear to students how even the best selected syllabi cannot claim to speak for the entire discipline nor be completely representative, let alone comprehensive or exhaustive. To this end, the conceptual framework could introduce methods and approaches which emphasise contextualised and situated knowledges, recognising that knowledge is place and time specific. Knowledge is underpinned by powers which have legitimised it as knowledge, often at the expense of other/alternative knowledges.

4. **Teach responsible resource extraction.** Emphasis should be placed on ethical, sustainable extraction and exploration. Cultural considerations should be embedded into our curricula (e.g. land ownership works in very different ways around the globe). Curricula should encourage students to explore where the majority of material extraction occurs vs. the abundance of the material globally. Explore what local environmental and human rights look like and compare the price of commodities and where those materials are being consumed. Case studies of local and Indigenous knowledge systems can be used to explore equitable partnerships with local communities. Examples of where Indigenous land, knowledge and culture has been destroyed can be used to frame these discussions.

5. **Explain the unethical practice of "parachute science" and unethical specimen extraction, to avoid the pitfall of extractivism**. Teach case studies and acknowledge how these events have negatively affected locals whilst benefiting the individuals/groups responsible. Explore how collaboration and co-creation with local groups would have led to benefits to all involved. Cultural ethical considerations should be embedded in research project design materials, from dissertation level to grant applications. Those with responsibilities of writing grant applications or leading field courses should be encouraged to account for working with local groups. True partnerships should be encouraged rather than Global North (senior) partners setting the research agenda and designing the project, and then inviting Global South partners on board. Ensure results are disseminated within the local community and in a form which can be assessed easily and is useful.

6. **Explore the bias of Global North research (abundance, 'impact' and perception).** There is a bias in both the number of papers produced by teams from the Global North - even where this research is focusing on topics from the Global South (North et al., 2020), and in the 'impact' and perception of the quality of work produced by researchers from the Global North vs. South (Collyer, 2018). Commit to including works from a broader range of

authors. Embed decolonised actions into research procedures – work with local researchers and people. Consider inviting researchers from the Global South for reviews and to provide virtual research seminars to students.

7. **Participate in creating a more diverse population of geologists.** Research has shown that projects run by diverse groups are more impactful (used more widely and cited more) than those with non-diverse project teams (AlShebli et al., 2018). This is also true of curricula, particularly those co-created with student bodies. Alternative knowledge can be offered and integrated within the curricula to appeal to a wider audience and resonate with a greater number of non-geologists as well as providing a broader range of knowledge systems, approaches and attitudes. Work towards dismantling hierarchies and structures that create barriers and exclude groups. Diverse representation likely creates more inclusive communities of practice (Sheffield, et al., 2021). A diverse body of geologists, including Indigenous scholars, is needed to tackle the grand challenges of the twenty-first century (Dowey et. al., 2021)

8. **Teach climate change as a social justice and colonial issue**. Geological knowledge of climate change is essential to understanding the dangers of the anthropogenically enhanced Climate Crisis. Teach students that climate change is not apolitical; it is an example of modern colonialism, with largest anthropogenic contribution to pollution from the Global North whilst the largest impacts are felt in the Global South (e.g. Weizman and Sheikh, 2015; Mahony and Enfield, 2018). Policy and process must be created with researchers and populations from the Global South to ensure equity of proposals and partnerships.

9. **Co-create and collaborate.** Traditional curricula tend to focus on the individual – whether that is in highlighting 'lone geniuses' or in emphasising individual academic achievement. Our curricula should emphasise the benefit of group and teamwork (Gregory and Thorly, 2013; Johnson and Johnson, 2009 and Springer et al., 1999) – including working with those with Indigenous and local knowledge bases. Students should be encouraged to create content or design parts of the curriculum (a choice in assessment style, for example). The exploration of knowledge and its creation should be encouraged and individuals should be steered towards processes that benefit them and those around them. Industry partners should be sought to help create authentic assessments based on complex issues and problems, and the human element of geological activities should be embedded alongside the physical and process-based narrative. Throughout Decolonising the Curriculum initiatives, students are our co-creators.

10. **Educate for sustainability.** Sustainability themes, issues and challenges are excellently curated into the United Nations Sustainable Development Goals (SDGs). Embedding exercises based around the SDGs is a useful way of examining how vital geology is to a sustainable future (e.g. Stewart, 2016; Stewart and Gill, 2017; Gill and Smith, 2021; Geology for Global Development, 2021), and how crucial it is to acknowledge colonial damage caused by geological activities. The SDGs can be tracked to activities across the geology curriculum (Rogers et al., 2018).

These actions are by no means exhaustive but aim to provide a starting point for geology academic teams beginning to think about Decolonising the Curriculum.

Sundberg (2014) highlights the importance of taking steps – moving, engaging, reflecting – in enacting decolonising practice, "understanding that decolonisation is something to be aspired to and enacted rather than a state of being that may be claimed". Sundberg encourages those undertaking decolonisation to progress by recognising and encompassing other forms of knowledge ('multiplicity'). They argue that we each create our own truth or knowledge, because we are all subject to different conditions; our experience of the world is not inevitable ('historical contingency'). This goes some way to explain why we find different knowledge in different societies and places; our lived experiences differ and so therefore does the way we build knowledge around these experiences. Historical contingency should not be a concept unfamiliar to geologists. The idea that historic (geological) events are not inevitable, but that each event relies on a number of complex conditions, is one that anyone reconstructing past Earth events will understand.

## 5.1 The Power of Decolonisation

Decolonising the Curriculum may initially feel inaccessible to scientists, with its own set of terminology/jargon and its basis in historical context. However, it is vital to a more equitable future for geology and many other disciplines, with value to both academics and students. It also serves as a reminder that the work we conduct is not apolitical, neutral, nor divorced from society – people, places, knowledge, power and the environment are interwoven with our science.

Decolonising any curriculum involves not just the contents of the syllabi, but the pedagogical structures underpinning the curriculum, from delivery right through to assessment methods. Decolonising the Curriculum is a set of processes, a pedagogical approach as well as an ideology, which seeks to enhance knowledge and learning, to make disciplines richer and more enthralling. It seeks to include more, to dig deeper, to encompass more viewpoints and representations and voices, to welcome diversity rather than stay narrow and limited. Decolonising is a democratic and collaborative process, breaking down hierarchies to heighten productivity and effectiveness. It talks truth to power, exposing power structures that have shored up practices and processes unseen and uncalled out for most of their existence. Decolonising curricula, if done well, should be a liberating process and an education enhancer for both staff and students.

## 6 Decolonising the Curriculum Glossary and Recommended Reading

In this resource we have tried to steer clear of language that may be unfamiliar, impenetrable or off-putting to many geologists (and probably individuals from other STEM subjects); or else, if we have used such terms, we have tried to explain them along the way. In this section, we highlight some key terms typically found in Decolonising the Curriculum literature in an attempt at de-mystifying them for those unfamiliar:

*Baconian knowledge*: Modern scientific method as developed by Sir Francis Bacon; a method of knowledge creation based on systematic observation resulting in empirical data.

*Colonial/colonialism*: the act, practice or policy of control of a people by a power or other people. Often associated with the establishment of colonies. Colonialism, the creation of colonies and their exploitation in systematised ways, derives from the Latin term 'colonia', which meant a settlement of Roman citizens in a newly conquered territory.

*Decolonisation*: entails the removal of ongoing colonial domination (Noxolo et al., 2017); in its early 20C usage, referred to the process of attaining political independence by colonised countries (the socioeconomic, historical and geographical act of nations gaining independence from colonisers). Decolonisation is often thought to be the end territorial ownership of colonies, however, colonialism does not disappear with decolonisation; coloniality can continue long after a country has decolonised.

*Epistemicide*: the systematic destruction of existing (usually of an Indigenous) knowledge base (Bennett, 2007)

*Epistemology*: How knowledge was produced.

*Epistemological violence*: where empirical data is interpreted in a way that implies the Other is inferior (Teo, 2010). Epistemic violence can be either *against* knowledge, or *through/via* knowledge (I.e. dominant knowledges, oppressive knowledges).

*Extractivism*: The process of extracting natural resources for export for economic/academic gain (often associated with poor environmental process and policy).

*Imperial/Imperialism*: relating to an empire and/or activities of an empire

*Firsting*: Knowledge (of a discovery, a finding etc.) framed from a European perspective for phenomenon that was made by Others previously. This framework of knowledge promotes (mostly white, male) Europeans as creators of global knowledge often to the detriment of those who are not accepted as "firsters" (Beck, 2017).

*Neoliberalism*: a movement with commitments to individual liberties, belief in shifts in policy and ideology against government intervention and a conviction that market forces should be self-regulating (Olssen et al., 2004)

*Neocolonial*: the process and/or practice of using economic and cultural influence and globalisation to influence or control a country, society or people, rather than through direct occupation and colonial governmental control. "Neo-colonialism is... the worst form of imperialism. For those who practise it, it means power without responsibility and for those who suffer from it, it means exploitation without redress." (Kwame Nkrumah, Ghana's first post-independence President, *Neo-Colonialism: The Last Stage of Imperialism.* London: Thomas Nelson and Sons, 1965.)

*Ontology*: What knowledge actually is as knowledge.

*Others, Othering, the Other*: individuals or groups who are presented as not fitting with the social norms of a social group. A process which influences how people view and treat the others and leads to in- and out-groups (Held, 2020).

*Pedagogy*: the interaction between students, teachers, the learning environment and learning activities (Murphy, 1996).

*Postcolonialism*: the study of the socioeconomic, historical, cultural and political impacts of colonialism on colonised people and their lands. N/B 'Post'colonialism does *not* mean 'after' colonialism; postcolonialism begins *at the first moment of colonial contact* and describes what transpires thereafter as a result of colonising or having been colonised.

*Privilege*: how a person's identity can afford them (often unacknowledged) advantages as a function of the group with which they identify. For example, social class, age, nationality, disability, ethnic or racial category, gender, neurodiversity, sexual orientation, and religion. One marker of privilege is that the privileged person need not even be consciously aware of that privilege and even when aware, can neglect the awareness through a kind of 'sanctioned ignorance'; whereas the absence of privilege throws up obstacles, problems, struggles and suffering which cannot be ignored.

We recommend the following texts/resources for those who wish to explore the theme of Decolonising the Curriculum further (there are many more relevant resources available, these are just some that we found particularly useful):

**Decolonising Curricula and Pedagogy in Higher Education: Bringing Decolonial Theory into Contact with Teaching Practice** (ThirdWorlds) by Shannon Morreira, Kathy Luckett, et al.

**Towards Decolonising the University: A Kaleidoscope for Empowered Action** by Dave S.P. Thomas and Jivraj Suhraiya

**Decolonising Intercultural Education: Colonial differences, the geopolitics of knowledge, and inter-epistemic dialogue** (Routledge Research in International and Comparative Education) by Robert Aman

**Dismantling Race in Higher Education: Racism, Whiteness and Decolonising the Academy** by Jason Arday and Heidi Safia Mirza

**Re-imagining Curriculum: Spaces for disruption** by Lynn Quinn

**Decolonising the University: The Challenge of Deep Cognitive Justice** by Boaventura de Sousa Santos

**Decolonizing Geography: an introduction** by Sarah Radcliffe

**Recognising Geology's Colonial History for Better Policy Today** by Maddy Nyblade and Jenn McDonald (see references)

**Decolonising the Curriculum** by Amrita Narang, York University (https://edta.info.yorku.ca/decolonizing-the-curriculum/)

**Decolonising higher education: creating space for southern sociologies of emergence** by Catherine Manathunga (see
references)

**Bryn Mawr Geology and Colonialism Reading List** (http://mineralogy.digital.brynmawr.edu/blog/geology-colonialism-reading-list/)

### Acknowledgements

The authors wish to thank the two referees and the editor of this paper who provided constructive, thought provoking and encouraging feedback, ultimately greatly improving this manuscript.

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
