# Peer review of "Geology Uprooted! Decolonising the Curriculum for Geologists."

_Geoscience Communication, 2021_

## Author Comment (AC1)

Dear referee,

Many thanks for the time you have taken reading and commenting on this submission. Both reviews we have received are incredibly constructive and informative and we deeply appreciate the effort that has been put into them. We generally agree with the feedback and suggestions made in both reviews. We do feel that some of the suggestions are out of scope for this manuscript and it's intended audience (people with no background to decolonising and similar concepts who are from a discipline heavily divorced from its human and social impact). This manuscript is designed as an introductory piece, and we understand that it often skirts around some complex arguments and concepts – and may feel lacking in depth to those who are familiar with the topics and concepts covered. Several of the suggestions made are things some of the authors (and other groups in the geosciences) are working towards and that we hope will be outputs in their own right. Reading these reviews has been hugely interesting and insightful, again we extend our thanks to the reviewers, hopefully our paths may cross again as we work towards a more inclusive, accessible and decolonised geology! In order to make our response more focused we have provided comment (in red) directly to each point:

My perspective on this ms is informed by my training in human geography, and most recently by my extensive work to write about decolonizing geography, including on physical geography. In this respect, I read the ms as an interested academic in a cognate discipline who has extensive knowledge of the decolonising debates generally.

Overall, the paper raises many important points about the presence of colonial legacies in geology's teaching, interpretive frameworks, and its canonisation of particular (western-endorsed) forms of knowledge. The ms is written in an accessible style in order to create a broad conversation, and without assuming any prior knowledge of decolonial issues. My main response is positive, although the ms could, I suggest, do more to draw out specific dimensions of the discipline's "colonial present" in more detail, as the discussion was at points rather general.

Colonial present was something we discussed in detail whilst putting the ms together. We did begin to write up a section focusing on it and it was quickly apparent that it would be a very significant section (a manuscript in its own right). It was felt that a large section on the colonial present in this ms may disengage the very individuals who we want to reach (decolonisation is uncomfortable and we need the discipline to understand that before those feelings create disengagement). There is current work being undertaken to look in more detail at the disciplines colonial past/collecting Indigenous geologies/outlining the colonial present (for example, several of the authors of this are part of a project that has recieved some funding for a project looking closer at the colonial present of the discipline).

p.1 "early modern" - be careful with this label which generally refers to the period 1492 to around 1700; so this ms's focus on the post-1700 period could more usefully be labelled as such. On the same page, "our civilization" is for many people a loaded term, both because it assumes a common heritage and the term has such strong associations with western societies. Perhaps "diverse world societies" would work?

Actioned – really interesting point about labeling of time periods (something geologists love doing!)

p.2, line 46: exploitation of mineral resources was undertaken by Spanish and Portuguese state-led colonisation and colonialism, from the early 16th century. line 50: these forms of knowledge may not have been 'academic' in the sense today, but they did establish a mindset that Europeans had the right to identify, extract and trade minerals, which were the foundations at a deeper level for later 'academic' study. Line 56: North European Empires (versus earlier southern European empires, including Spain and Portugal) - clarification required here. And it would be useful to explain to non-specialist readers what is meant here by "dominance of knowledge production" - what is described here is epistemology [note spelling; incorrectly spelled at end of ms] which then informs pedagogy (students tend to be inducted into the dominant epistemology, through a particular mode of teaching and learning ie pedagogy). Decolonising the curriculum entails questioning and reforming both epistemology (the interpretive frameworks, and domains of what is considered important knowledge) and pedagogy.

Can clarify on Empires. We avoided the use of non-specialist terminology as much as possible but we agree this would be a good place to expand on epistemology (and point the reader to the glossary), will also add pedagogy to the glossary.

p.3 line 69: provide a page number for Peake and Kobayashi. lines 73-75: in decolonising debates today, the emphasis has been on the colonial geopolitics (ie which world powers, which world scientific networks and associations) of knowledge production as a whole. It is this pattern of power and domination that then shapes which types of knowledge are deemed valuable enough to 'extract' or destroy. So I wondered if more could be said here about the norms, 'standards' (established in geology) and control of journals, and who decides what is valuable 'new' knowledge. Yes, journals as an example would be a useful addition here – and an example the target audience will be familiar with. Line 81: students 'learn' or 'engage with' (the term 'undertakes' is awkward). In this paragraph, the crucial point is that the curriculum is endorsed as the knowledge most worth passing on - if there is a very common set of truths found across all geology courses, then that can be termed the 'canon'. Deciding what goes in/stays out of the canon relates back to the power dynamics mentioned above. Could the ms give an example or two of geology's canon? Examples of what isn't canon might be easier? Social Justice - Ethics - human facing content - is often missing as it is dismissed as being outside of the discipline? Line 93: decolonising the curriculum is not solely concerned with repositioning theory then, but also crucially the content and - key, although underexamined in the ms currently - the interpretive frameworks used to explain and understand the content. Not too sure many geologists would be aware of interpretive frameworks, but I think a sentence around theory, content and pedagogy would be useful.

p.4 line 100 the phrase "acknowledge colonial debts to knowledge creation" is awkward - arguably, it is a question of colonial legacies in knowledge creation. Actioned

p.5 line 140 One aspect requiring some more discussion is the unacknowledged influence of particular types of geologists relative to others in setting norms and criteria for excellence in a discipline - so it is a mindset and an awareness of the issues that need to change. The ms in this respect could perhaps delve more into the reasons why STEM subjects present their knowledge as 'neutral' and unaffected by social relations (again it's not a question of individuals but of society-wide 'commonsense'). This is important as decolonizing is not about just adding in Indigenous peoples and stirring; not least as the dividing line between Indigenous and STEM knowledges are context specific, blurred, contested and very difficult to fix. Good point, we can add some discussion around this. Lines 150-55: it would be very useful here to give an extended example to illustrate how an interpretation could be repositioned in a decolonising way - eg. from a first year undergraduate lecture, or topic that everyone learns in university; this would offer a specific set of resources as well as a sustained argument about how to decolonise [the paper does end with a list of recommendations, but these are not particularly engaged with the specifics of geology, which is what readers will presumably want to find here.] Since this page contains a considerable amount of repetition, a specific example would move the paper along strongly here. There are plans for a much more detailed output that introduces exactly this, by introducing this as a separate output we would hope to be able to provide a much greater/detailed set of resources/case studies/methods/ideas.

p.6 'colonised geology' : this subsection could usefully start with a line or two about the purpose of the historic overview - is this not taught in geology courses? If it is taught on courses, it is rare! Certainly, in the UK – History of Geology modules are aligned to Earth (geological) history, rather than the history and origin of the subject itself.  Also, the discussion is very Anglo focused, and underplays the global nature of geology's history. I recommend looking at J A Secord 2018 chapter 'Global geology and the tectonics of empire', in Curry et al (eds) Worlds of Natural History. And A Bobbette and A Donovan (eds.) 2020 Political Geology. Springer. And Nicola Miller 2020 Chapter 7: Land and Territory, in her book Republics of Knowledge, Princeton University Press. These give a more international sense of how colonialism and geological knowledges interacted, drawing out German, Spanish and other trajectories. Many thanks for these suggestions, we can work them in. Though Anglo focused is important as it seems to be predominantly Anglo activity that bought about the current canon (with other European nations working within this Anglo "framework"/set of knowledge).

pp.6-7: this account tends to reinforce the idea that white European men are the problem (with a few notable exceptions); rather than this, how about talking about the type of exploration, the systems of validation, and institutional standards and circuits of knowledge production, by which core chronologies, typologies, international systems for verification etc etc were set up and - likely - continue to have a resounding influence today. In other words, structures not people. This is an interesting point. We can certainly look to include more emphasis on the systems and procedures inherent in the past and present discipline. We do think

that focusing on the human element is important in this manuscript, some geologists struggle to see the human impact of the discipline and of their own actions, showing the human element behind the structures created which persist into the present subject is important (discussions around core chronologies, typologies, circuits of knowledge production and systems of validation are likely to be out of place for the target audience?)

p.8 line 230: what are the global norms at play today? Could this discussion be more specific? Again, it's not just a question of random Indigenous peoples currently outside western institutions; decolonising implies a mindset in which a variety of knowledges - from farmers, women, Black, non-dominant religious  groups etc etc - could pluralise and diversify what geology means. Another way to come at this question is to ask what is it that seems to make Indigenous knowledges so incompatible with 'global' geology? We can expand on this – we have tried to emphasise the point that it's not a question of integrating indigenous peoples into western institutions with some of the examples used in the section pg6-7, and call attention to the fact that geological knowledge is plentiful outside of the present discipline canon. The idea of pluralising geology is interesting, this might be a useful way of talking about a discipline that isn't based on the current canon?

line 236/38: on culture and science distinction: decolonial analysis highlights how non-western societies are said to have 'culture' while western societies have 'science'. We can change our use of "cultural framework" here as this just emphasises its colonialism – and also explain non-western 'culture' vs western 'science'.

line 260-4: underscore the power of geologists relative to the legal provision for specific people and places; this reflects unequal power, not just geopolitics over national legislation, but crucially the role of political and economic elites and powerful groups, who ally themselves with western geologists (especially if it means foreign earnings from mineral wealth). Line 266: surely this is geology's "colonial present" (Gregory 2004) not its past? Absolutely – should read present and decol actions needs to read in the future.

p.10 line 306 the word is "willingness" Actioned

p.10-.11 - see Bryn Mawr college website for an extensive list of resources on the relationship between colonialism and geology, including links to Indigenous policy to establish ethical entry into and prospecting on Indigenous lands and territories.

http://mineralogy.digital.brynmawr.edu/blog/geology-colonialism-reading-list/

We are aware of this collection, some of the recourses will be impenetrable to some geologists (which is why we didn't include it) but we will add it to the suggested reading list – it is a very useful collection.

p.11 line 355: what does "more impactful" mean in this context? Could this be unpacked more.

line 361: on the Anthropocene, there is an emerging decolonising debate: see Davis H and Todd Z 2017 On the importance of a date, or decolonizing the Anthropocene. ACME 16(4):

761-80. And Slaymaker, O, Mulrennan, M and Catto N. 2020 Implications of the Anthropocene epoch in geomorphology, In Landscapes and Landforms in Eastern Canada. Springer.

It is probably best to avoid detailed debates around the Anthropocene here – lots of opinions in geology. It is interesting that it originated as a geological time period (and often referred to as such where it used) but most definitions do not use geological events to frame it.

p.13 spelling: epistemology, and epistemicide - you may be interested in Boaventura de Sousa Santos who has written extensively about epistemicide from a critical social science perspective. Actioned

It is great to see a glossary of terms given here! Another resource, in due course, will be S Radcliffe Decolonizing Geography: an introduction, Polity Press, Cambridge UK (published April 2022 in UK, May 2022 in North America).

Actioned, a resource from such a cognate discipline will be a useful addition.

Again, many thanks for this review and the effort put into it.

---

## Author Comment (AC2)

Dear referee,

Many thanks for the time you have taken reading and commenting on this submission. Both reviews we have received are incredibly constructive and informative and we deeply appreciate the effort that has been put into them. We generally agree with the feedback and suggestions made in both reviews. We do feel that some of the suggestions are out of scope for this manuscript and it's intended audience (people with no background to decolonising and similar concepts who are from a discipline heavily divorced from its human and social impact). This manuscript is designed as an introductory piece, and we understand that it often skirts around some complex arguments and concepts – and may feel lacking in depth to those who are familiar with the topics and concepts covered. Several of the suggestions made are things some of the authors (and other groups in the geosciences) are working towards and that we hope will be outputs in their own right. Reading these reviews has been hugely interesting and insightful, again we extend our thanks to the reviewers, hopefully our paths may cross again as we work towards a more inclusive, accessible and decolonised geology! In order to make our response more focused we have provided comment (in red) directly to each point:

Dear Author(s),

I enjoyed reading this article and appreciated the politics that lead you to write this work. In particular, your call to make accessible to the STEM community (and very specifically Geologists) the importance of "decolonizing the curriculum" is noteworthy. You organize the article well and set out to outline how the field of Geology is deeply grounded in colonized knowledge production mechanisms, and the impact this continues to have on the current field. You focus on the 'decolonizing the curriculum' as a site from which you can imagine a new field – that it cognizant of its history, but also willing to make the change required to ensure that the knowledge produced is inclusive, accessible, and diverse. This seems quite laudable goals, and you end your writing with concrete ways in which the field of geology (with sustained efforts by current practicing and teaching geologists) can change. You provide 10 concrete ways to do this (with also a focus on climate justice as part of the 10 points suggested). The writing ends with a glossary, which I think non-specialists will find particularly useful.

While this is a well-intentioned article and writing, I'm afraid it also has a few vital flaws, which I outline below as way to possible encourage the author(s) to rethink/rewrite/reframe this writing.

1. A collapse between decolonial/post-colonial/anti-colonial frameworks – This seems like a vital issue with your writing. Decolonizing as a political term comes from a long legacy of Indigenous scholars working to ensure that Indigenous knowledge

and ways of knowing are recognized as vital ways of organizing our world – BUT it is also vitally about the land on which settlers live and thrive (including the University). Decolonizing work is then different from scholars who do post-colonial work and scholarship. While you cite Tuck and Young (Decolonization is not a metaphor), there is no engagement with his scholarship – nor an attempt to resolve how decolonizing the curriculum engages with the larger politics of decolonization and IS NOT a metaphor (or is it?).  A non-critical engagement with 'decolonizing the curriculum' is another form of privilege that you as authors need to interrogate – and then build your own analysis from for your future facing geology projects.

We have tried to frame this paper specifically around Decolonising the Curriculum as this is an initiative many academics are being asked/encouraged to engage with but don't understand what it is. We completely acknowledge that the depth and rigor of the framing of Decolonial/post-colonial /anti-colonial frameworks is not what may be necessarily expected in a manuscript exploring these topics (particularly by scholars who are familiar with the topics!) – the manuscript is written to be an "entry" guide into Decolonising the Curriculum for geologists, which explains the outline concepts (with some historical context/examples/why it matters included), highlights false narratives (e.g. removal or "cancelling" of content/individuals) and importantly emphasises the human element of the discipline. Talking about decolonisation as a (e.g.) metaphor (or not) would potentially disengage readers and reinforce the (anecdotally at least) prominent thinking amongst many geologists that decolonisation is "something for the Humanities".

2. Please remember, that diversity is NOT decolonization. I recommend a few bits of easy reading to help clarify this vital point in your writing:

3. http://www.criticalethnicstudiesjournal.org/blog/2019/1/21/do-not-decolonize-if-you-are-not-decolonizing-alternate-language-to-navigate-desires-for-progressive-academia-6y5sg
4. https://speakingofmedicine.plos.org/2021/07/29/its-time-to-decolonize-the-decolonization-movement/
5. https://aninjusticemag.com/its-not-decolonize-it-s-desupremify-9b6e9ea02aae

We understand this and do not intend to suggest that diversity = decolonisation. We have highlighted that diversification of materials (etc.) is NOT decolonising the Curriculum (and why).

Point 7 of our suggestions does specifically call for diverse geologists. This is because geology (particularly postgraduate level) has a huge diversity problem, dominated by white men. This structure is colonial and inherent of the current disciplines colonial origin. We acknowledge that diversifying geology doesn't necessarily equate to decolonisation, but a diverse set of voices are needed to shape any decolonisation of the discipline.

6. Open your article with the ways geology existed in pre-colonial/indigenous knowledge spaces: Currently in your writing, the structure situates "geology" proper as a science, as a field of knowledge – as a way of knowing the world. While you acknowledge this 'formation' came about because of colonization, you reinforce this privileging of 'Science' with a capital S by situating it above localized knowledge(s). Even as you set to situate Geology, you begin by citing Nicholas Steno. I recommend moving up section 4.2 first, and then outlining the ways geological knowledge existed in certain locations – and what colonization did to erasures of these knowledge(s) and ways of knowing the world. In your own writing, highlight the erasures and violence of colonization. This ensures that the scholars who read your work are willing to recognize explicitly the violence of their academic ancestors, and understand that geological knowledge existed long before it was formalized within and through colonial science (as an aside, you can critically engage with the science must fall framework – I do not sign up for that entirely, but it goes help me think through some vital points critically).

We have structured the manuscript in the order it is in to ensure readers recognise the discipline that they "know" and can then introduce how/why it is colonial – then moving on to examples of what some of these structures are. We feel that many of the scholars we are trying to engage with would likely disengage if erasure and violence of the discipline was upfront and explicit (no matter how true it is). We wholly agree that a detailed and honest account of pre-colonial and Indigenous geology is required, with an account of how this has been erased/stolen. To this end several of the authors of this manuscript, along with historians, and decolonial scholars have successfully applied for (some modest) funding to begin pursuing this. This is the logical next stage of resources aimed at geologists to continue highlighting the colonial past/present of the discipline but requires additional outputs.

7. Of the 10 recommendations you have for your field, point 7 is about the diversity of scholars in the field. However, at no point do you engage with a systematic analysis of how many Indigenous scholars are Indigenous – either in your own universities, nationally, or withing the leading Geology organizations. Is there no research on this – and if not, perhaps that is a gap you can address. Inclusion in 'teams' can sometimes be tokenistic. However, mapping out how many professors of Geology are Indigenous scholars or how many recent hires are Indigenous junior scholars might be a concrete way to highlight how the field of Geology continuous to be a settler colonial field – with ongoing violence both on the lands of the people on which the research is done and where it is taught (i.e. the physical space the Universities stand on).

Several of the authors are part of a group of geologists who are actively working to increase the inclusivity, accessibility and diversity of geology – it is all too clear that geological academia in the Global North is massively white and male. We are not aware of any data on how many geology scholars are

Indigenous scholars but looking at "general" diversity statistics would indicate not many are – it is certainly a colonial field of study (particularly once postgrad levels are reached). We will emphasise the importance of working with, and alongside, Indigenous scholars and populations. In geology there are many colonial, inclusivity, accessibility and diversity issues, we need to ensure that the burden of work to remove these barriers does not fall heavily upon those who have been historically marginalised and/or had their knowledge erased or stolen (which is one of the reasons for writing this sort of manuscript).

8. Situate yourself within this writing: This might be harder for you to do, as in the STEM fields we still want to believe in the ideas of 'objective' knowledge – when, our best bet is to work with 'situated knowledges.' Sandra Harding's work is truly helpful in this framing and given that you already draw on their work – I would encourage you to develop this a bit more (there is a recent sage research chapter on Sandra Harding's work would work well for your STEM audiences). It would be helpful for the readers to know how many of you are Indigenous scholars, and the experiences you may have within the field of geology. Also, maybe concretely outlining how you bring your ancestral knowledge to bear on 'traditional' geology curriculums.

We can explore a way to introduce the authors backgrounds and experiences – however this needs to be thought about carefully, we know that there are individuals openly hostile to the ideas of decolonising the geology curriculum (amongst other problematic issues the discipline has) and would not wish to provide any information that could lead to harassment.

In conclusion, I commend the authors for this work and encourage them to consider re-framing this article, so it sets out to fulfil its own political goals. I'd also encourage them to work closely with Indigenous scholars (which is different from working with/in diverse research teams). In your conclusion, you beautifully remind your readers that geologists need to remember that "[…] work we conduct is not apolitical, neutral, nor divorced from society – people, places, knowledge, power and the environment are interwoven with our science." Yes, indeed!!!

A pleasure to engage with this work and I wish the authors well in their pursuit to shape Geology for the next generation.

Again – many thanks for this review. We appreciate that the manuscript does not wholly delve into many of the colonial issues in geology, or fully introduce the epistemocide that lays the foundations for the present canon of the discipline. These are important, and we need to work towards these goals. This manuscript could be seen as one of the first steps into introducing the disciplines problematic past and present, decolonisation, and Decolonising the Curriculum, to a discipline of scholars who for the most part are unaware (or unwilling to except) the human impact of geology.

---

## Author Response (AR1)

Dear Rebecca,

We would first like to thank you for your time and effort in handling the editorial process for this submission so far, and for your comments which have provided constructive guidance for our revisions.

We have taken on board the comments of both referees, and those made by yourself and have made some substantial additions/changes to the manuscript. The main changes introduce being: 1) a more global picture of geology's colonial past (by including activities of the southern European colonial powers and introducing their role in laying the foundations for the modern geological canon); 2) a more global picture of decolonising the curriculum and its context within wider decolonial theory; 3) more engagement with some of the foundational concepts/debates of decolonisation and colonialism – we hope here that we have provided plenty for interested readers to get to grips with whilst keeping the manuscript accessible to readers who are simply interested in finding out what Decolonising the Curriculum is (most probably including a fair share of "sceptics"!). By providing a suggested reading list we hope that those readers who are interested in learning more about decol have a ready set of resources with which they can do so.

Providing a line-by-line account of the revisions we have made would probably take longer to read then the manuscript itself(!) so we have compiled a list of point-by-point responses to your editor comments (in red below) and linked these to examples of where we have actioned the suggestions (the page and line numbers given correspond to the Track Changes version – where all the revisions can be seen).

With regard to Reviewer 1:

There are a few topics reviewer 1 asks you to expand on that you say you don't have space for here, but that you will be writing future papers on. Can you mention that in this paper somewhere? Eg at the end of the introduction, or at the end of the paper, you could say, 'future papers will focus on …' to let readers know that you are aware of these issues and there is more to come (eg colonial present, case studies).

We have included sentences which better explain this manuscripts framing (e.g. as a scene setting/introductory step – and have contextualised decolonising the geology curriculum within the much larger decolonisation discourse) – and highlighted some of the areas that the authors, our collaborators etc., are working on or towards (e.g., pg 2 ln 39-43; pg2-3 ln 51-68). Hopefully some of the readers might contribute to these areas too!

With regard to the reviewer's comment on 'interpretive frameworks' I think this is a self-explanatory term and safe to use.

I want to respond to your comment on the teaching of the history of geology being rare – I certainly remember being taught the history of geology in New Zealand in the late 1980s! It possibly was mostly covered in my physical geography/geomorphology classes, rather than my geology classes, but just a reminder that your audience for this paper is international, not just geology academics in the UK, and there will be breadth to their own experiences.

We have expanded the introduction to this section to outline that there is some history of geology taught in most courses – but that it generally focuses on the contributions of certain individuals, and rarely the systems and circumstances under which those contributions are made. The section on historical geology has been moved after the discussion on what Decolonising the Curriculum is about. (Pg, 11 ln 349-355)

With regard to your query as to 'discussions around core chronologies, typologies, circuits of knowledge production etc being out of place for the target audience' – I would urge you not to underestimate your audience. If you can find a way to include this in the paper, even a paragraph, I think this would improve the paper and an understanding of this could be revelatory for your audience.

We have included several areas expanding on the more theoretical discussions of decolonisation and what is and isn't canon – and who decides that - but we have tried to keep the discussion on this introductory in line with the scope of the piece (we have expanded the glossary explanations in places to ensure that everyone should have the info to unravel unfamiliar terms) (e.g. Pg 4 ln 124-130; pg 5 ln 134-147; pg 6 ln 179-182; pg 7 ln 194-198; pg8 ln 249-254)

The reviewer made a comment about p11, line 355 that you've not responded to.

Sorry – that one slipped through! We have explained "more impactful" as the authors of the cited paper did; they essentially claim citations = impact (we have extended this a little to include general use as we do not agree that citation = impact...) We have also included indigenous scholars and broader knowledge systems to this section (pg 18 ln 579-584)

I agree that you can leave the Anthropocene out of this paper.

This would make a very interesting paper, possibly one for the future!

Definitely include Decolonizing Geography in your reading list, it might also be worth finding out if you can see an advance copy of Radcliffe's book, given the crossover between geology and physical geography there might be something useful here for you.

Included, and looking forward to it arriving in the library!

With regard to Reviewer 2:

1. I think you're doing your readers a disservice by saying some of these topics will disengage them. Please consider putting some of this more challenging material into its own section, so that those who are up for this more sort of material can read it? The geologists I work with are up for it and want this paper, and I wouldn't worry about challenging them.

There are definitely geologists who are up for the challenge, but there are also definitely those who are unsure or openly sceptical/hostile to initiatives like decolonisation – we hope that this revised version can cater for both groups – and that for those coming away wanting to know more, the recommended reading should hopefully provide a good place to start. We have included several additional references highlighting some recent work in this area, which may also be useful for geologists looking to explore further.

6. Again, you think readers will disengage ... why? Don't underestimate your readers.

We have made some significant changes based on this paragraph of feedback, including moving what was 4.2 to be 4.1 (and vice versa). Several examples are included where we highlight that the current canon exists because of the wilful ignorance, erasure or theft of indigenous knowledge – and this is an area that some of the authors are working in to highlight some of these stories in some detail. We have included some examples of indigenous and local geological knowledge to highlight that these exist, and we have tried to question why this knowledge is at "odds" with geological canon. We have included examples of knowledge that isn't typically considered geological canon (politics for example – pg 8 ln 249-255)

8. I endorse the idea to bring your background into the piece but of course would not want you to do anything that might lead to a risk of harassment? But I'm also surprised by this comment. Your names are already on the paper, so anyone who wants to know more about you can look each of you up online. It would be helpful to know something about the authors, and would be an honest and transparent thing to do to put this in the paper, and relevant to this topic. If you don't want to identify individuals, you could outline your backgrounds as part of a list – from the info on the paper I can see that you are five scholars who are based in the UK. Are you all geologists? Or are some from other disciplines? Are any of you indigenous – if so to what country? No need to say who is who if you put all the information together like this (I've done similar things in papers that I've co-authored).

Your suggestion of a list is a welcome one – we certainly agree that situating ourselves and our relationship with knowledge production is important. We have also included why we have written such a piece. (But yes, sadly there are quite a few cases of some reasonably aggressive "push back" to discourse suggesting that geology has negative parts, particularly on social media...) (Pg 2 ln 45-53)

As a general comment, as you undertake your revision I would urge you not to be too defensive: rather than thinking of your audience as geologists likely to be resistant to the topic, inclined to harassment, write for the geologists who are hungry for this information, want to engage with this topic, and need some guidance on how to go about it. Don't let people who are resistent to change define what this paper could be. And also remember this paper is for an international audience and the people likely to find and read it are likely to be the people who are already beginning to be engaged in this topic and want guidance.

Hopefully this revised version is much more geared towards those wanting to engage rather than those who might be resistant (whilst being of potential use to both groups – the hope is that the former group might be able to use it to help inform/engage the latter). The addition of some more examples of geologies colonial past (particularly Spanish/Portuguese Empire expansion) (eg pg 3 ln 86-91; pg 13 ln 392-402), the recognition of settler colonial nations tending to be "further along" the decol journey (with examples/references) and better explanation of coloniality, postcolonial decolonisation hopefully better caters for an international audience – and most certainly provides a good platform for further discussion around Decolonising the Curriculum in postcolonial vs external colonial nations vs settler colonial nations, and provides a foundation for further exploration of specific international geological knowledge systems. (e.g. pg 5 ln 155-158; pg 10 ln 303-312)

We feel that the revisions have greatly improved the manuscript, so again we extend our thanks to the reviewers and yourself for your constructive comments. We look forward to hearing from you.

Best wishes, the author team.

---

## Author Response (AR2)

Dear Rebecca,

Many thanks for these corrections – all of which have been actioned. The references have been checked and revised where needed.

Many thanks for the handling of this paper at all stages.

Best wishes,

The author team

---

## Author Response (AR3)

Authors response FINAL

In this version references have been reviewed to conform with the author guidelines – this was mostly moving the year of publication to the correct place, and including https:// at the stard of DOIs